# CP Merging: Joint LoRA Merging using Canonical Polyadic Decomposition

## Abstract

Large language models (LLMs) are often fine-tuned for specific tasks using Low-Rank Adaptation (LoRA), an efficient method that adds small, task-specific modules called LoRA adapters to a pre-trained base model. However, a major challenge arises when merging multiple LoRA adapters trained on different data sources for a specific task: it often leads to *task interference*, which refers to the redundancy or sign discrepancies found in parameters across different task models, resulting in information conflict and performance loss. SVD-based, such as TSV merging, attempt to reduce the task interference by leveraging the geometric structure of the singular task matrix, which may prevent the identification of the optimal orthogonal basis and task updates. This leads to a question: *Does a higher-dimensional task representation merging improve multi-task performance by providing a more robust basis for task-specific singular components?* To answer this, we propose a tensor-theoretic approach using CP decomposition to lift task-specific parameters into a higher-order manifold, thereby exploiting the uniqueness properties of tensor factorizations to minimize task interference. By aggregating adapters into a third-order tensor, our method leverages the unique identifiability of CP factors to disentangle shared knowledge from task-specific features. This joint factorization naturally mitigates cross-task interference by capturing the underlying multi-linear structure of the LoRA adapters. Our extensive experiments further validate this approach, demonstrating that CP merging yields superior performance compared to existing SVD-based merging approaches.

## 1 Introduction

Parameter-efficient fine-tuning (PEFT) methods have emerged as practical alternatives to full fine-tuning for adapting large language models (LLMs) to downstream tasks (Hu et al., 2021; Houlsby et al., 2019b; Lester et al., 2021). Among these, LoRA adapters (Hu et al., 2021) have gained particular popularity by learning low-rank additive updates that efficiently capture task-specific knowledge. Recent studies (Sheng et al., 2023; Ostapenko et al., 2024; Huang et al., 2023) have shown that LoRA can be fine-tuned for diverse tasks—such as summarization, question answering, and classification—resulting in a set of pre-trained LoRA adapters (LoRA library).

The emergence of model hubs, such as Hugging Face, has further simplified the sharing and reuse of pre-trained and fine-tuned models. These platforms enable collaborative and multi-task learning by allowing users to easily acquire and extend existing models. Within this context, model merging—which aims to combine multiple specialized models into a single model capable of handling diverse tasks—has attracted significant attention (Wortsman et al., 2022; Yadav et al., 2023; Yang et al., 2026; Stoica et al., 2025; Gargiulo et al., 2025). However, merging task-specific models often introduces task interference, since parameter updates optimized for one task may conflict with those optimized for another (Yadav et al., 2023). To mitigate this issue, a variety of model merging methods have been proposed (Davari & Belilovsky, 2024; Ilharco et al., 2022; Matena & Raffel, 2022; Wortsman et al., 2022; Yadav et al., 2023; Yu et al., 2024; Lu et al., 2024; Yang et al., 2026; Deep et al., 2024; Miyano & Arase, 2025; Chen et al., 2025). However, traditional methods such as Fisher Merging or TIES-Merging operate on the full parameter space of the model. These methods typically focus on resolving conflicts by calculating weight-level importance or trimming "redundant" parameters based

on their magnitude or Fisher information. While effective for fully fine-tuned models, they assume that all weights contribute equally to the underlying representation. Unlike full-model merging, merging LoRA adapters involves combining highly compressed, low-rank matrices ($BA$) that reside in specific subspaces of the original model. The challenge here is not just weight consensus, but subspace alignment. In addition, Prior work has observed that LoRA adapter weights exhibit weaker alignment compared to fully fine-tuned model weights (Stoica et al., 2025). By comparing the pairwise centered kernel alignment (CKA) Kornblith et al. (2019) between full models and their equivalent LoRA finetuned counterparts. They observe the full models have very high CKA (the finetuning-updates have high alignment for task updates) while LoRA exhibits considerably lower CKA.

Consequently, applying existing model merging techniques directly to LoRAs often yields suboptimal results. To address this, researchers have explored LoRA-specific merging strategies, such as singular value decomposition (SVD)–based methods. TSV merging Gargiulo et al. (2025) attempt to reduce the task interference by building the *singular task vector* using SVD. The overlapping singular vectors suggest shared features in the weight space across tasks, which can introduce interference when models are merged. Then they explicitly address task interference by leveraging the geometric structure of singular vectors to minimize interactions between task-specific parameters. Knots Stoica et al. (2025) build upon the SVD to transfer the task updates of different LoRA models into a shared space. They first concatenate all task-updates for a layer, and then decompose the result with the SVD to obtain $U\Sigma V^T$, U are singular vectors that form the *orthonormal basis* for a shared representation between all tasks. The non-negative diagonal values in $\Sigma$ represent the scale associated with each basis vector, and then the Knots can apply the existing merging methods to the $V^T$ task-matrices and obtain a $V^{(merged)}$. Despite the success of these SVD-based methods, these SVD-based merging approaches (e.g., `TSV-merging`, `Knots`) treat the LoRA adapters merely as matrices, which may prevent the identification of the optimal orthogonal basis and task updates, and as a result, limit their ability to effectively reduce cross-task interference. So there leads a natural question: *Does a higher-dimensional task representation merging improve multi-task performance by providing a more robust basis for task-specific singular components?*

To answer this question, we treat the collection of LoRA adapters as a unified structure by concatenating them into a third-order tensor and then performing tensor decomposition. Formally, given $N$ tasks, each associated with a LoRA adapter $\Delta_i \in \mathbb{R}^{d_{in} \times d_{out}}$, we concatenate all adapters to construct a third-order tensor. Unlike prior approaches that apply independent SVD-based decompositions to mitigate task interference (Stoica et al., 2025; Gargiulo et al., 2025), we employ Canonical Polyadic (CP) decomposition to jointly factorize all task-specific matrices. CP decomposition expresses a tensor as a sum of rank-one components, thereby disentangling shared and task-specific patterns. This joint factorization not only helps reduce task interference but also preserves task-relevant information more effectively. Finally, we merge the LoRA adapters by summing over the task dimension.

To rigorously evaluate our CP merging approach, we conduct experiments on 10 held-in tasks and 3 held-out multi-tasks from the Flan datasets (Based on Phi-3 3B) **?**, as well as on 10 out-of-domain downstream tasks (Based on Phi-3 3B and Mistral-7B) Ostapenko et al. (2024) and skill-composition tasks (Based on LLama 7B) Prabhakar et al. (2024). For 10 held-in Flan tasks, CP merging outperforms strong SVD-based baselines by +3.19 Rouge-L. For the held-out Flan tasks, CP merging achieves an improvement of +5.69 Rouge-L over the SVD baseline. Interestingly, we observe that uniform merging yields the best performance among all methods on the held-out evaluation. For the 10 out-of-domain (zero-shot) tasks, CP merging surpasses SVD baselines by an average of +1.0%, while uniform merging also shows competitive performance in this setting. Finally, in the skill-composition evaluation, CP merging outperforms SVD-based methods by +2.0% accuracy on the challenging math-hard task.

In summary, our contributions are:

- We propose a CP merging approach that mitigates task interference in LoRA merging by jointly modeling shared and task-specific components in a unified manner.

- Experimental results show that CP merging substantially improves 10 held-in performance (+3.19 Rouge-L), 3 held-out tasks (+5.69 Rouge-L), and skill composition tasks (2% accuracy) over SVD-

based methods. This demonstrates that CP decomposition effectively disentangles task-specific factors from shared factors, thereby reducing interference while preserving essential task information.

## 2 Related Works

**PEFT**. Parameter-efficient fine-tuning (PEFT) methods facilitate efficient adaptation of LLMs without updating all the training parameters, thereby reducing the memory and computation cost (Houlsby et al., 2019a; Zhang et al., 2024; Zaken et al., 2022; Guo et al., 2021; Li & Liang, 2021; Lester et al., 2021). Hu et al. (2021) proposes a method named LoRA that parameterizes incremental weights $\Delta$ as a low-rank matrix by the product of the down projector matrix and up projector matrix. Zhang et al. (2023) proposes Adaptive Low-Rank Adaptation (AdaLoRA), a new method that dynamically allocates the parameter budget among weight matrices during LoRA-like fine-tuning (Zhang et al., 2023). Liu et al. (2024) introduces a new approach DoRA to investigate the inherent differences between full fine-tuning and LoRA.

**LoRA library**. In a multi-task scenario, LoRA adapters can be fine-tuned for diverse tasks-resulting in a set of pre-trained LoRA libraries. AdapterSoup Chronopoulou et al. (2023) trains each adapter for each domain and performs weight-space averaging of adapters trained on different domains. Huang et al. (2023) introduces LoRAhub to aggregate the LoRA modules trained on diverse tasks. AdapterFusion (Pfeiffer et al., 2021) proposes a two-stage algorithm that leverages knowledge from multiple tasks. Similar to LoRAhub, a group of task-specific adapters learn to encapsulate the task-specific information, and in the second stage, a fusion layer combines the trained adapters. Ostapenko et al. (2024) propose a model-based clustering library building methods and an Arrow routing function to reuse the LoRA library. Zhao et al. (2024) propose a retrieval-augmented mixture of LoRA Experts (RAMoLE) that adaptively retrieves and composes multiple LoRAs according to the input prompts.

**Model Merging**. Model merging integrates the weights of multiple task-specific models into a single multi-task model (Davari & Belilovsky, 2024; Ilharco et al., 2022; Matena & Raffel, 2022; Wortsman et al., 2022; Yadav et al., 2023; Yu et al., 2024; Lu et al., 2024; Yang et al., 2026; Deep et al., 2024; Miyano & Arase, 2025; Chen et al., 2025). Early methods like Model Soups Wortsman et al. (2022); Choi et al. (2024) average the weights of multiple models fine-tuned with different hyperparameters. Matena & Raffel (2022) `Fisher Merging` proposes to use the Fisher information matrix to merge the various models. Ilharco et al. (2022) presents the `task vectors`, which are the weight differences between pretrained models and the finetuned models. Then they merge the task vectors to obtain a merged multi-task model instead of the whole model. To reduce the task interference during merging, `Ties` Yadav et al. (2023) reduces the parameter redundancy by selecting the top-$k$ most significant parameters and then constructing a sign vector based on the majority sign. `DARE` Yu et al. (2024) randomly drop delta parameters with a ratio $p$ and rescales the remain parameters by $1/(1-p)$ to reduce the task interference. `RegMean, RegMean++` Jin et al.; Nguyen et al. (2025) addresses the model merging by minimizing prediction differences between the merged model and the expert model. `TSV` Gargiulo et al. (2025) shows that SVD decomposition can reduce the parameter redundancy, and merging the singular values can compress the parameters and reduce the task interference. `Knots` Stoica et al. (2025) uses the SVD to jointly transform the weights of different LoRA models into a shared space. `ISO-C` Marczak et al. (2025) propose an isotropic merging framework that flattens the isotropic merging framework to flatten singular value spectrum of the task matrix, enhancing alignment and reducing task interference. Instead of decomposing the task matrix separately, we concatenate the task matrix into a third-order tensor and then decompose it using CP decomposition.

## 3 Preliminaries

### 3.1 Pre-trained Models and LoRA Parameterization

Let $W_{\text{base}} \in \mathbb{R}^{d_{\text{out}} \times d_{\text{in}}}$ denote the pre-trained weight matrix for a given layer of a neural network. When adapting the model to a downstream task, Low-Rank Adaptation (LoRA) injects trainable rank-decomposition matrices while keeping $W_{\text{base}}$ frozen. The weight update is parameterized as $\Delta W = BA$, where $B \in \mathbb{R}^{d_{\text{out}} \times r}$ and $A \in \mathbb{R}^{r \times d_{\text{in}}}$, with the rank constrained such that $r \ll \min(d_{\text{in}}, d_{\text{out}})$. The forward pass for an input feature $x \in \mathbb{R}^{d_{\text{in}}}$ is thus modified to $y = (W_{\text{base}} + \alpha \Delta W)x$, where $\alpha$ is the scaling factor of the LoRA adapter.

For notational simplicity, we omit layer-specific indices. However, all the following definitions and merging operations are applied layer-wise across the entire architecture.

## 3.2 Gradient-Free Model Merging

We consider a scenario where a single pre-trained model is independently fine-tuned on $K$ distinct tasks using LoRA. This yields a pool of $K$ task-specific models. For any given layer, the weights of the $k$-th fine-tuned model are defined as:

$$W_k = W_{\text{base}} + \alpha \Delta W_k$$

where $\Delta W_k$ represents the task-specific LoRA update learned during the fine-tuning of task $k$.

Our objective is to fuse the parameters of the $K$ fine-tuned models into a single, unified model capable of solving all tasks. We focus on gradient-free approaches, which are highly data-efficient and enable on-the-fly merging without requiring access to the original training data. Existing gradient-free methods merge models by linearly combining the task-specific updates. For a given layer, the weight matrix of the merged model is computed as:

$$W_{\text{merged}} = W_{\text{base}} + \alpha \sum_{k=1}^{K} \Delta W_k$$

## 3.3 CP Decomposition

To analyze the interplay between multiple LoRA adapters, we leverage the framework of tensor decomposition. Tensor decomposition aims to approximate a tensor through a set of low-rank factors with diverse applications. We specifically focus on the Canonical Polyadic Decomposition (CPD), which was originally introduced by Hitchcock (1927; 1928) as a method to represent a tensor as a sum of a minimum number of rank-one tensors. CPD (Fig. 1)is often considered a special case of the more general Tucker Decomposition (De Lathauwer et al., 2000; Tucker, 1966), which decomposes a tensor into a core tensor multiplied by a factor matrix along each mode. While Tucker decomposition offers high flexibility, we utilize CPD in this work because its uniqueness properties—under mild con-

Figure 1: An illustration of SVD decomposition of a matrix $A \in \mathbb{R}^{m \times n}$ and CP decomposition of third-order tensor $\mathcal{T}$.

ditions—allow for the identifiable isolation of task-specific information, which is critical for mitigating task interference. In this way, a high-order tensor can be uniquely represented as the sum of a set of rank-one tensors. Given a third-order tensor $\mathcal{T} \in \mathbb{R}^{d \times r \times N}$, the CP decomposition is expressed as:

$$\mathcal{T} = \sum_{i=1}^{R} \lambda_i \mathbf{u}_i \otimes \mathbf{v}_i \otimes \mathbf{s}_i \tag{1}$$

where $R$ is the rank of CP decomposition. $\lambda_i$ is the scaling weight. $\mathbf{u}_i$, $\mathbf{v}_i$ and $\mathbf{s}_i$ are the factor vectors corresponding to the three modes of $\mathcal{T}$. The operator $\otimes$ denotes the outer product.

## 4 Method

### 4.1 CP Merging

**Task Singular Vectors** (Gargiulo et al., 2025) use SVD to decompose the layer task matrices and term the obtained singular vectors *Task Singular Vectors (TSV)*, revealing low-rank properties and deeper insight into the inter-task interactions. Given two tasks $i$ and $j$, the tasks matrix $\Delta_i$ and $\Delta_j$ can written as:

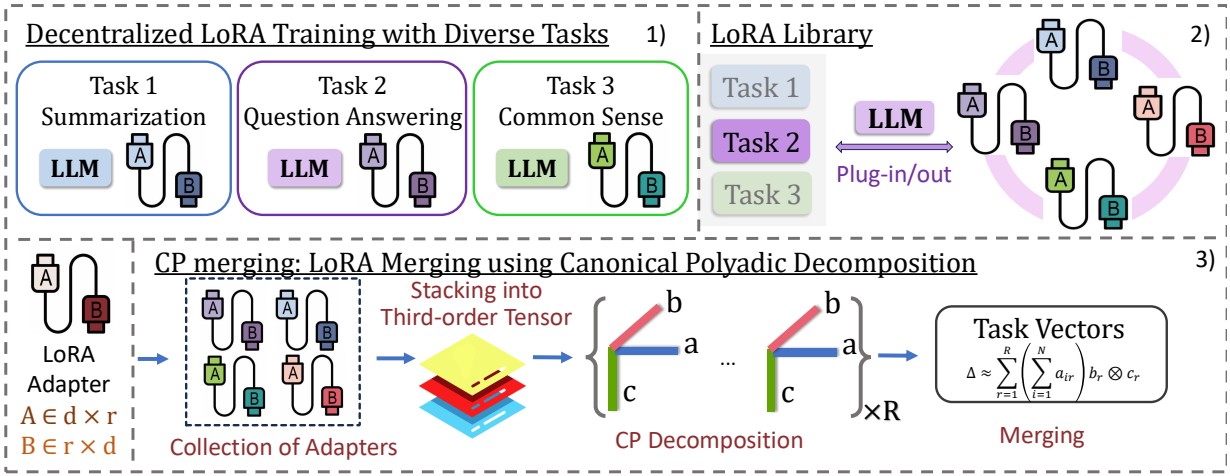

Figure 2: 1) LoRA adapter from various tasks, each LoRA adapter is trained individually. 2) LoRA library aims to leverage the plug-in/out nature of LoRAs to offer the ability to add or remove knowledge from LLMs. 3) CP merging: we stack the LoRA into a third-order tensor and conduct the CP merging.

$$\Delta_i = U_i \Sigma_i V_i^\top, \quad \Delta_j = U_j \Sigma_j V_j^\top \tag{2}$$

where $U_i$, $U_j$ are the left singular vectors and $V_i$, $V_j$ are the right singular vectors. $\Sigma_i, \Sigma_j$ are diagonal matrices of singular values. As shown in Fig. 1, we can interpret SVD as a sum of rank-one matrices. For each task $i$, we get the best approximation of each task matrix $\Delta_i$ by retaining only the top-$k$ singular values and their corresponding vectors:

$$\hat{\Delta}_i = \sum_{j=1}^{k} \sigma_j^i \, \mathbf{u}_j^i \otimes \mathbf{v}_j^i. \tag{3}$$

where $\hat{\Delta}_i$ is the low-rank approximation of the task matrix for task $i$, obtained using the top-$k$ singular values $\sigma_{i,j}$, left singular vectors $\mathbf{u}_{i,j}$ and right singular vectors $\mathbf{v}_{i,j}$. `Knots` conduct the SVD decomposition individually to obtain the right singular matrix $[V_1, V_2, ..., V_N]$ and then merge the right singular matrix to obtain a merged matrix $V^{(\text{merged})}$. The final task matrix can be computed by: $DeltaW_j^{(\text{merged})} = U\Sigma[V^{(\text{merged})}]^T$. `TSV merging` decomposes the LoRA adapters individually and then concatenates the left singular matrix $[U_1, U_2, ..., U_N]$ and right singular matrix $[V_1, V_2, ..., V_N]$. We notice that SVD decomposes the task adapter separately, leading to independent choices of SVD dimensions. To address this, we propose a more general global decomposition using Canonical Polyadic (CP) decomposition, which takes all the LoRAs together as a tensor input.

---

**Algorithm 1** Implementation of CP Merging

1: **Input:** Pretrained task LoRA adapters $\{\Delta_1, \Delta_2, ..., \Delta_N\}$, scaling factors $\alpha$.
2: **Output: Merged weight** $W_{\text{MT}}$
3: **Concatenate** the matrices:
4: $\mathcal{T} \leftarrow [\Delta_1|\Delta_2|...|\Delta_N]$
5: **CP Decomposition:**
6: $\mathcal{T} \approx \sum_{r=1}^{R} \mathbf{a}_r \otimes \mathbf{b}_r \otimes \mathbf{c}_r$ (Eq. 4)
7: **Compute the Task Matrix:**
8: $\Delta_i \approx \sum_{r=1}^{R} a_{ir} \mathbf{b}_r \otimes \mathbf{c}_r$ (Eq.5)
9: **Reconstruct** the merged matrix:
10: $\Delta \approx \sum_{r=1}^{R} \left( \sum_{i=1}^{N} a_{ir} \right) \mathbf{b}_r \otimes \mathbf{c}_r$ (Eq. 7)
11: **Construct** the merged model weights:
12: $W_{\text{merged}} = W_{\text{base}} + \alpha \sum_{r=1}^{R} \left( \sum_{i=1}^{N} a_{ir} \right) \mathbf{b}_r \otimes \mathbf{c}_r$
13: **return** $W_{\text{MT}}$

**Representing all adapters as a Third-Order Tensor** We consider the task adapters $\Delta_i$ for all tasks $i$ as slices of a third-order tensor $\mathcal{T}$, where the dimensions could be interpreted as (tasks, input dimension, output dimension). Let use denote this tensor $\mathcal{T} \in \mathbb{R}^{d_{in} \times d_{out} \times N}$ where $N$ is the number of tasks, $d_{in}, d_{out}$ are the dimensions of each task matrix $\Delta_i$. Each $\Delta_i$ (a matrix of size $d_{in} \times d_{out}$) is a frontal slice of $\mathcal{T}$ along the task mode. So, we can concatenate the $\Delta_i$ matrices along the task dimension to form $\mathcal{T}$.

**CP Decomposition Form** In CP decomposition, the tensor $\mathcal{T}$ is approximated as a sum of rank-one tensors:

$$\mathcal{T} \approx \sum_{r=1}^{R} \lambda_i \mathbf{a}_r \otimes \mathbf{b}_r \otimes \mathbf{c}_r \tag{4}$$

where $R$ is the number of components (analogous to $k$ in SVD), $\mathbf{a}_r \in \mathbb{R}^N$ is a vector associated with the task mode, $\mathbf{b}_r \in \mathbb{R}^d_{in}$ and $\mathbf{c}_r \in \mathbb{R}^d_{out}$ are vectors associated with the row and column modes. In our paper, we set $\lambda_i = 1$, For each task $i$, the $i$-th slice $\Delta_i$ of $\mathcal{T}$ would be:

$$\Delta_i \approx \sum_{r=1}^{R} a_{ir} \mathbf{b}_r \otimes \mathbf{c}_r \tag{5}$$

where $a_{ir}$ is the $i$-th element of $\mathbf{a}_r$, selecting the contribution of the $r$-th component for task $i$.

**CP merging over the task mode** To merge the $\Delta_i$ into a final $\Delta$, we sum over the task dimension (mode-1). The merged $\Delta$ would be:

$$\Delta = \sum_{i=1}^{N} \Delta_i \approx \sum_{i=1}^{N} \sum_{r=1}^{R} a_{ir} \mathbf{b}_r \otimes \mathbf{c}_r \tag{6}$$

Since the outer product $\mathbf{b}_r \otimes \mathbf{c}_r$ is the same for all $i$, and assuming $a_{ir}$ represents the weight of component $r$ for task $i$, we can factorize this as:

$$\Delta \approx \sum_{r=1}^{R} \left( \sum_{i=1}^{N} a_{ir} \right) \mathbf{b}_r \otimes \mathbf{c}_r \tag{7}$$

where $\sum_{i=1}^{N} a_{ir}$ is the total contribution of the $r$-th component across all tasks. The final merged weights can be written as :

$$W_{\text{MT}} = W_0 + \sum_{r=1}^{R} \left( \sum_{i=1}^{N} a_{ir} \right) \mathbf{b}_r \otimes \mathbf{c}_r \tag{8}$$

## 5    Experimental Results

To test the effectiveness of our approach, we evaluate CP Merging on a variety of LoRA merging methods in held-in and held-out multi-tasks based on Flan datasets (Sec. 5.1)Arnob et al.; Ostapenko et al. (2024) and zero-shot task (Sec. 5.2)Ostapenko et al. (2024) and skill composition tasks (Appendix A.2)Prabhakar et al. (2025).

**Baselines** We compare the following methods in the zero-shot benchmark. **BASE**: The base model without adaptation. **Multi-task**: A single expert LoRA finetuned on a joint training set. **Uniform**: For each task, we train a LoRA adapter, then we merge the LoRA uniformly (Huang et al., 2023; Chronopoulou et al., 2023). **Task Arithmetic (TA)** (Ilharco et al., 2022) subtracts the parameter values of the pre-trained

model from those of the fine-tuned models, creating a set of "task-vectors." Then they linearly summed to create a merged model. **Ties-merging** (Yadav et al., 2023): Ties improves TA by resolving the parameter interference between models when merging. It prunes low-magnitude weights and then only averages the weights that share the dominant sign. **DARE** randomly drops fine-tuned weights and rescales the remaining ones to create sparse task vectors Yu et al. (2024). **TSV merging** uses SVD decomposition to compress the LoRA and reduce the task interference based on the singular vectors (Marczak et al., 2025). **Knots** uses the SVD to transform the weights of jointly different LoRA models into a shared space Stoica et al. (2025). **ISO-C** Marczak et al. (2025) proposes an isotropic merging framework that flattens the singular value spectrum of the task matrix.

| Model | Wiqa | Sciq | Adver | Duorc | Cos | Wikiqa | Quail | Wikihop | ParaphQa | Yelp | AVG | 3 Held-Out |
|---|---|---|---|---|---|---|---|---|---|---|---|---|
| **Phi-3 (3B) with LoRA library** | | | | | | | | | | | | |
| BASE | 14.55 | 7.99 | 10.85 | 7.32 | 5.00 | 4.17 | 11.88 | 3.51 | 6.93 | 8.33 | 8.05 | 16.00 |
| Oracle | 36.87 | 79.56 | 51.58 | 52.40 | 42.99 | 55.94 | 51.51 | 46.58 | 8.325 | 32.71 | 45.85 | - |
| Uniform | 22.98 | 61.97 | 29.61 | 18.45 | 28.12 | 11.56 | 38.94 | 7.194 | 7.18 | 15.66 | 24.17 | **61.13** |
| Ties-merging | 23.33 | 51.52 | 30.15 | 18.05 | 30.12 | 12.54 | 35.05 | 10.03 | 5.80 | 16.76 | 23.34 | 34.69 |
| Ties w/DARE | 25.42 | 52.76 | 32.34 | 19.04 | 31.20 | 13.33 | 36.05 | 7.05 | 6.08 | 16.00 | 23.93 | 34.70 |
| ISO-C | 14.59 | 8.19 | 11.11 | 7.50 | 5.38 | 4.20 | 12.23 | 3.53 | 6.95 | 8.39 | 8.20 | 16.18 |
| TSV Merging | 36.75 | 53.52 | 44.39 | 20.20 | 59.13 | 23.20 | 37.25 | 34.70 | 4.78 | 34.95 | 34.89 | 45.70 |
| Knots | 26.33 | 90.81 | 40.15 | 28.87 | 50.44 | 48.35 | 43.24 | 13.71 | 6.63 | 23.56 | 37.21 | 54.46 |
| CP Merging | 34.69 | 89.88 | 44.32 | 31.59 | 66.82 | 50.34 | 45.49 | 10.73 | 5.90 | 24.23 | **40.40** | 60.15 |

Table 1: 10 held-in and 3 held-out Flan tasks based on Phi-3 3B model. We report the Rouge-L scores. The oracle here denotes that we use the trained LoRA to test the corresponding tasks, which achieve the best results for held-in results. The task details are presented in Appendix A.4.3.

## 5.1 Held-In and Held-out Multi-Tasks

We conduct our experiments using the FLAN dataset (Longpre et al., 2023) and sample 10 held-in and 3 held-out tasks following (Ostapenko et al., 2024; **?**). Each task is sub-sampled to 10,000 examples. Within these samples, 1,000 are allocated for validation and early stopping.

As shown in Tab. 1. The CP Merging method achieves the highest average score of 40.40, indicating it generally performs best across the tested datasets. It also has a strong 3-Held-Out score of 60.15. CP Merging is the standout performer, showing significant gains over other methods, especially in the average score and the 3-Held-Out evaluation. The BASE model provides a baseline performance, while the Oracle provides an upper bound, representing the theoretical maximum performance without any merging techniques. Ties-merging and Ties w/DARE show similar average performance (23-24). ISO-C has a significantly lower average score, similar to the BASE model's performance. TSV Merging and Knots show moderate performance, with Knots

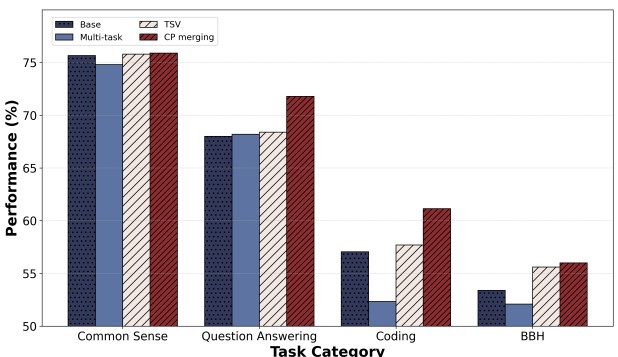

Figure 3: Zero-shot results across different categories. The X-axis represents the task categories.

having a higher average score and a much higher 3-Held-Out score. In short, the CP Merging method appears to be the most effective overall, while Uniform merging excels on the specific 3-Held-Out evaluation.

## 5.2 Zero-shot Experiments

To test the out-of-domain generalization, we evaluate CP Merging on a variety of zero-shot task benchmarks. Specifically, we experiment with four broad classes of zero-shot tasks: (1) **Common Sense Reasoning**: WinoGrande (Sakaguchi et al., 2021), HellaSwag (Zellers et al., 2019), and PIQA (Bisk et al., 2020). 2)

| | Common Sense | | | Question Answering | | | | Coding | | Reasoning | AVG |
|---|---|---|---|---|---|---|---|---|---|---|---|
| | Piqa | Wg | Hswag | Boolq | Obqa | ArcE | ArcC | HE | Mbpp | BBH | AVG |
| Phi-3 (3B) with LoRA library | | | | | | | | | | | |
| BASE | 81.1 | 70.2 | 75.7 | 85.0 | **49.0** | 80.0 | 57.9 | 53.0 | 61.1 | 53.4 | 66.6 |
| Multi-task | 79.1 | 69.6 | 75.6 | 87.1 | 47.6 | 82.6 | 55.6 | 51.8 | 52.9 | 52.1 | 65.4 |
| Uniform | 81.0 | 70.4 | 76.0 | 84.7 | 48.8 | 82.4 | 57.9 | 52.4 | 63.0 | 55.6 | 67.2 |
| Ties-merging | 80.8 | **70.9** | 75.4 | 80.8 | 46.0 | 85.8 | 60.8 | 54.9 | 61.5 | 51.9 | 66.9 |
| Task Arithmetic | 81.0 | 70.5 | 75.9 | 84.7 | 48.8 | 82.5 | 58.2 | 54.3 | 61.9 | 54.2 | 67.2 |
| Ties w/DARE | 80.7 | 70.8 | 75.2 | 82.8 | 46.3 | 84.8 | 60.9 | 54.9 | 61.6 | 52.0 | 67.0 |
| TA w/DARE | 81.2 | 70.1 | 75.3 | 84.9 | 48.9 | 82.2 | 58.3 | 54.5 | 61.1 | 54.3 | 67.1 |
| TSV Merging | 81.1 | 70.8 | 75.8 | 86.0 | **49.0** | 84.3 | 60.4 | 55.0 | 61.2 | 54.0 | 67.7 |
| Phi-3 (3B) with C-LoRA library | | | | | | | | | | | |
| Uniform | 81.5 | 69.5 | 76.0 | 86.6 | 48.8 | 86.7 | 61.8 | 57.3 | 65.3 | 55.9 | 68.9 |
| Knots | 80.8 | 71.7 | 74.9 | 85.0 | 48.0 | 84.1 | 58.2 | 53.2 | 62.6 | 50.4 | 66.9 |
| TSV Merging | 81.3 | 69.3 | 75.6 | 86.4 | 48.8 | 86.1 | 62.3 | 52.4 | 63.3 | 55.1 | 68.1 |
| CP merging | **81.7** | 70.0 | **76.0** | **86.6** | 48.9 | **86.7** | **62.5** | **57.3** | **65.4** | **56.0** | **69.1** |

Table 2: 10 downstream Zero-shot results based on Phi-3 (microsoft/Phi-3-mini-4k-instruct). C-LoRA is a LoRA library based on embedding clustering (A.6). We show the results based on Mistral in Appendix A.7. The experimental setting is presented in Appendix A.4.

**Question Answering**: BoolQ Clark et al. (2019), OpenbookQA Mihaylov et al. (2018), ARC-easy Clark et al. (2018), and ARC-challenge Clark et al. (2018). 3) **Coding**: HumanEval Chen et al. (2021), MBPP Austin et al. (2021). 4) **General-Purpose Reasoning**: BBH (Suzgun et al., 2023).

### 5.2.1 Experimental results

Tab. 2 presents the average zero-shot accuracy results across 10 downstream tasks. The multi-task based on the Phi-3 model (65.4%) underperforms the base model (66.6%), indicating that the multi-task training may encounter *catastrophic forgetting* issue that newly learned parameters overwrite or interfere with knowledge acquired during prior training (Wang et al., 2024). LoRA library with uniform merging, consisting of 256 experts, achieves accuracies of 67.2% for Phi-3.

Compared to the C-LoRA library, TSV Merging and Knots are comparable in overall performance. TSV Merging has an average score of 68.1, slightly better than Knots' average of 66.9. While TSV Merging and Knots perform well in some tasks, they do not consistently achieve the top scores that CP merging does. For example, TSV Merging ties for the highest score in Obqa (49.0), but its overall performance is less dominant. CP merging consistently shows superior performance. It achieves the highest average score of 69.1, outperforming all other methods. The results clearly demonstrate the effectiveness of CP merging, which consistently outperforms the other methods, including Knots and TSV Merging. With the highest average score and leading performance in multiple specific tasks, CP merging shows a more robust and effective approach to merging LoRA adapters, successfully mitigating task interference and enhancing overall model performance.

We further analyze performance across different task categories (Fig. 3). CP merging improves over the LoRA library with TSV merging, with the largest gains observed in question answering and coding tasks. In contrast, improvements on common-sense reasoning are minimal, suggesting that the impact of task interference varies by task type.

## 6 Ablation Studies

### 6.1 Number of Tasks analysis

Overall, the merging methods significantly outperformed the Base model across all task sizes. The Base model consistently showed the lowest ROUGE-L scores, with scores of 11.14, 9.15, and 8.05 for 3, 5, and 10

tasks, respectively. This suggests that LoRA merging is an effective technique for improving performance. CP consistently delivered the best performance on ROUGE-L scores. For 5 tasks, TSV merging and Knots merging performed equally well, both achieving a score of 54.4. The CP merging method surpasses Knots merging with a score of 57.7 for this task size. For 10 tasks, both the Knots and TSV merging methods saw a considerable drop in performance. In contrast, the CP merging method demonstrated superior robustness to an increase in the number of tasks, achieving the highest score of 40.40 in this category. The results indicate that LoRA merging is a highly effective strategy for improving model performance on multiple tasks. Specifically, the CP merging method shows the most promise, as it consistently achieves top-tier performance and exhibits greater stability as the number of tasks increases.

| Model | 3 tasks | 5 tasks | 10 tasks |
|---|---|---|---|
| Base | 11.14 | 9.15 | 8.05 |
| TSV merging | 44.87 | 54.4 | 34.89 |
| Knots | 59.04 | 54.4 | 37.21 |
| CP merging | **59.06** | **57.7** | **40.40** |

Table 3: The influence of the task size for LoRA merging. We select 3, 5, and 10 tasks from the Flan datasets and report the average Rouge-L scores for all the tasks.

### 6.2 CP Merging for Task Interference

To study the task interference, Gargiulo et al. (2025) introduces a score of task interference (termed as *Singular Task Interference*) based on the interplay of TSVs from different tasks:

$$\mathbf{STI}\left(\{\Delta_i\}_{i=1}^N\right) = \left\|\left(U^\top U - I\right)\Sigma\left(V^\top V - I\right)\right\|_1 \tag{9}$$

They assume that higher inner product values for $U^T U$ and $V^T V$ imply a higher likelihood (Gargiulo et al., 2025). The intuition is that overlapping singular vectors suggest shared features in the weight space across tasks. Inspired by this, we can reformulate the STI with CP decomposition. To define a CP-based STI metric, we need to assess the interference caused by the interplay of these factor matrices across tasks. The intuition remains that overlapping components (shared features in the weight space) lead to interference when merged. We can adapt the STI by considering the alignment and orthogonality of the factor matrices $a_r$, $b_r$, and $c_r$ across tasks. Then we define the CP-based Singular Task Interference (CP-STI) as:

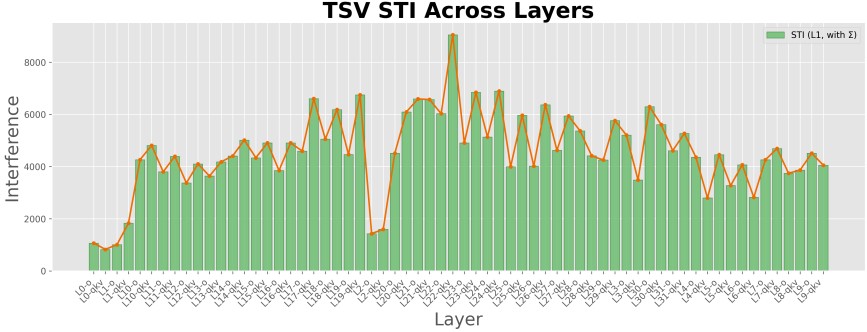

Figure 4: STI evaluation in TSV across layers based on 5 flan tasks.

where $A = [a_1, a_2, \ldots, a_R] \in \mathbb{R}^{N \times R}$ is the matrix formed by stacking the task-mode factor vectors $a_r$, $B = [b_1, b_2, \ldots, b_R] \in \mathbb{R}^{d \times R}$ is the matrix of row-mode factors, $C = [c_1, c_2, \ldots, c_R] \in \mathbb{R}^{d \times R}$ is the matrix of column-mode factors, $\circ$ denotes the Hadamard (element-wise) product, $I$ is the identity matrix of appropriate dimension (e.g., $R \times R$). Overlapping factors (high inner products) suggest shared features across tasks, which

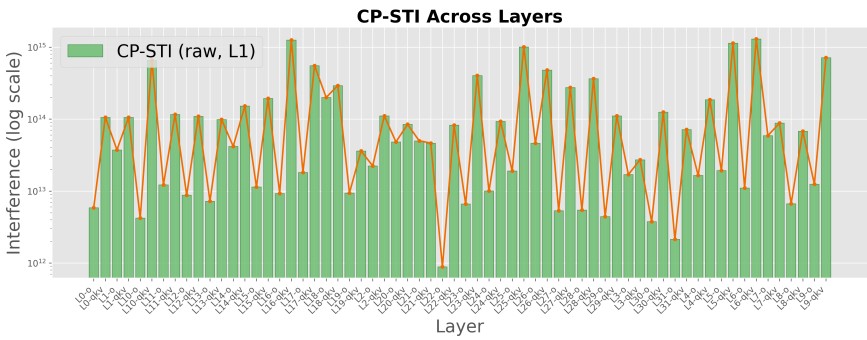

Figure 5: CP-STI evaluation in CP merging across layers based on 5 flan tasks.

| Metric | Statistic | Value | Interpretation |
|---|---|---|---|
| STI | Pearson(layer index, STI) | 0.755 | Positive linear trend across depth |
| STI | Spearman(layer index, STI) | 0.807 | Positive monotonic trend across depth |
| STI | mean(qkv_proj)/mean(o_proj) | 1.14x | Moderate projection asymmetry |
| CP-STI | Pearson(layer index, log CP-STI) | -0.006 | Near-zero depth trend |
| CP-STI | Spearman(layer index, log CP-STI) | -0.063 | Near-zero monotonic depth trend |
| CP-STI | mean(qkv_proj)/mean(o_proj) | 14.29x | Strong projection/magnitude asymmetry |
| CP-STI column-normalized | Pearson(layer, norm) | 0.306 | STI-comparable scale |
| CP-STI column-normalized | Spearman(layer, norm) | 0.324 | Moderate depth trend |
| CP-STI off-diagonal | Pearson(layer, offdiag) | 0.420 | |
| CP-STI off-diagonal | Spearman(layer, offdiag) | 0.533 | |
| Diagonal dominance | mean diag fraction | 1.000 | |

Table 4: Layer-wise interference statistics computed from per-layer TSV-STI and CP-STI. For depth correlation, layer-level scores are obtained by averaging o_proj and qkv_proj per layer. For CP-STI, log10 is used before correlation due to multi-order magnitude spread.

can introduce interference when the decomposed representations are merged, potentially degrading individual task performance.

$$\mathbf{CP} - \mathbf{STI}(\{\Delta_i\}_{i=1}^N) = \|(A^T A - I) \circ (B^T B - I) \circ (C^T C - I)\|_1 \tag{10}$$

As shown in the Figure 4, we visualize the STI metric across layers with five tasks based on Phi-3. We can notice that STI shows a clear depth thread. Aggregating over o-proj and qkv-proj, STI increases with layer depth (Pearson with layer index $\approx 0.755$, Spearman $\approx 0.807$), indicating stronger interference in deeper layers for this run. This conclusion does not match the conclusion int the TSV paper, where the interference is in the lower layer instead of higher layers. In addition, in STI, qkv-proj is consistently higher than o-proj (mean ratio $\approx=1.14x$), and top-interference points are mostly qkv-proj in deeper layers. (17, 19,22-24). As shown in Table 4, CP-STI is much larger on qkv-proj than o-proj (mean ratio=14.3x), showing strong layer/projection heterogeneity. Off-diagonal CP-STI has 0.53, and normalized CP-STI has 0.32, indicating the same direction of trend (deeper layers show stronger interference signals.

We also compared CP-STI computed with L1 and Frobenius norms in Figure 6. The key observation is that both norms produce highly consistent layer-wise patterns: peaks and valleys occur at the same layers in both panels. Frobenius values are typically lower in magnitude, but the relative ranking across layers is preserved,

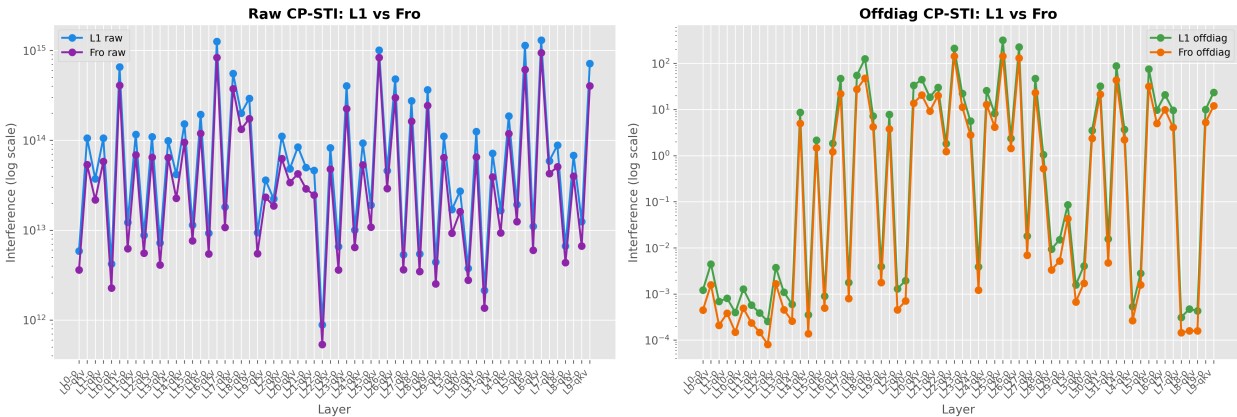

Figure 6: left: raw CP-STI, right: off-diagonal CP-STI.

indicating that the interference signal is not an artifact of a specific norm. Importantly, this consistency also holds for the off-diagonal variant: L1-offdiag and Fro-offdiag track each other closely across layers. Therefore, the metric's conclusion (where interference is stronger/weaker across layers) is norm-robust.

### 6.3 Rank $R$ of the CP decomposition.

CP decomposition factorizes a tensor into a sum of $R$ rank-one components. Wang et al. (2023) observed that increasing $R$ can improve performance on cross-modal tasks. We investigate how the CP rank affects our method's performance on the GSM8K-hard dataset (Fig.7). The left plot ("CP Rank vs. Accuracy") shows accuracy rising from 0.12 at rank 5 to a peak of 0.15 at rank 20, with no further gains beyond rank 25. This suggests that higher ranks capture richer patterns, but benefits plateau after rank 20. The right plot ("CP Rank vs. Invalid Code") shows invalid code outputs dropping from 340 at rank 5 to 200 at rank 20, with little improvement at higher ranks. Overall, increasing the CP rank improves both accuracy and output validity up to an optimal point (around rank 20), after which returns diminish, highlighting the importance of selecting an appropriate rank to balance performance and efficiency in mitigating task interference in skill composition tasks.

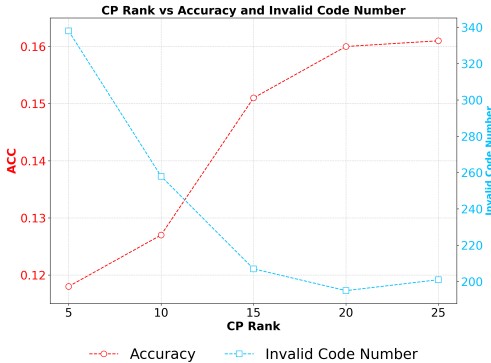

Figure 7: The impact of CP rank on model performance and generation quality for the GSM-8K-hard benchmark.

### 6.4 Alpha analysis

The alpha parameter effectively scales the influence of the adapter's output before it is added back to the weights of the original model. Table 8 reports the ROUGE-L scores for 10 held-in Flan tasks across seven $\alpha$ values ranging from 0.3 to 0.1. The results highlight the critical importance of the $\alpha$ hyperparameter for LoRA merging methods—including CP merging, knots, TSV merging, and ISO merging. Notably, the optimal $\alpha$ is not universal but depends on the specific task, underscoring the need for careful tuning. On average, moderate values (e.g., $\alpha = 0.15$–$0.18$) yield the best performance. For all experiments, $\alpha$ was selected based on performance over a validation

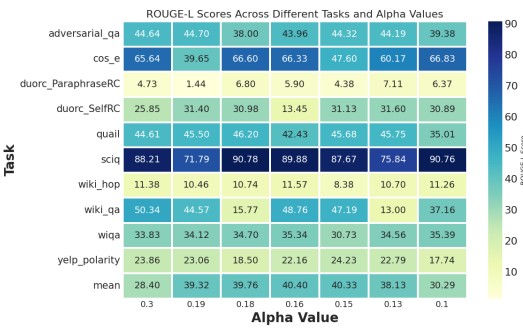

Figure 8: Best performance for different alpha values over 10 held-in Flan datasets.

| Model | 3 LoRA | 5 LoRA | 10 LoRA |
|---|---|---|---|
| TSV merging | 350s | 553s | 1320s |
| Knots | 540s | 780s | 1980s |
| CP merging | **31s** | **38s** | **58s** |

Table 5: Computation cost (seconds) compared with various LoRA merging approaches. We use Phi-3B as a backbone model.

set of 1,000 examples. Finally, we use 0.25 for TSV merging
and 0.50 for Knots.

### 6.5 Computation Cost

CP merging conducts the tensor decomposition, while the SVD-based merging conducts the matrix decomposition. In this subsection, we study the computation cost compared to these methods. We compare the computation cost for [3, 5, 10] LoRA adapters. The computation is based on the Phi-3B on Flan datasets. We run experiments using one A100 GPU card. As depicted in Tab.5, TSV merging and Knots exhibit substantial growth in cost as the number of LoRA modules increases, reaching up to 1320–1980 seconds with 10 LoRAs. In contrast, CP merging is significantly more efficient, requiring only 31–58 seconds across all settings—an order-of-magnitude improvement over the other baselines.

## 7 Discussion

The current formulation of CP merge assumes a static set of $N$ adapters merged into a third-order tensor $\mathcal{T} \in \mathbb{R}^{N \times d \times d}$. However, in dynamic "Open World" environments, new task-specific adapters are frequently released post-initial merging. A potential concern is the computational overhead of re-performing the full CP decomposition as the task dimension $N$ grows.

To address this, our framework can be extended to an incremental update strategy. Since the row-mode factor $B$ and column-mode factor $C$ capture the fundamental geometric structures of the weight space across layers, these can be treated as a fixed basis once the initial model is established. For a newly arriving adapter $\Delta_{N+1}$, instead of re-decomposing the entire tensor, we can solve a linear sub-problem to find a new task-mode vector $a_{N+1}$ that minimizes the reconstruction error:

$$\min_{a_{N+1}} \|\Delta_{N+1} - \sum_{r=1}^{R} a_{N+1,r}(b_r \circ c_r)\|_F \tag{11}$$

In this context, $a_{N+1}$ serves as the task coefficient, effectively embedding the new task into the pre-existing shared knowledge space without altering the established parameters of prior models. This approach significantly reduces the computational complexity from full tensor factorization to a simple least-squares projection, ensuring the engineering viability of our method in evolving multi-task systems.

## 8 Conclusion

We propose a novel LoRA merging method using Canonical Polyadic (CP) decomposition. Unlike existing SVD-based methods that decompose each LoRA adapter with matrix decomposition, CP merging factorizes multiple adapters in a unified manner. This allows us to disentangle task-specific components from those that are shared across tasks. Our comprehensive experiments show that CP merging consistently outperforms strong SVD-based baselines across various benchmarks, including in-domain multi-tasks, zero-shot tasks, and skill composition tasks. These results confirm that our method effectively preserves essential task knowledge while significantly reducing cross-task interference. This unified factorization approach is a key advantage

over conventional merging techniques. Our findings offer a new perspective on how knowledge is represented and merged within LoRA adapters.

### Acknowledgments

Use unnumbered third level headings for the acknowledgments. All acknowledgments, including those to funding agencies, go at the end of the paper. Only add this information once your submission is accepted and deanonymized.

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

# A Appendix

## A.1 Limitations

In this work, we evaluate our approach on natural language tasks using models such as Phi-3B, Mistral-7B, and LLaMA-7B. Extending the analysis to larger-scale models remains an important direction for future research. Our experiments show that CP merging achieves superior performance compared to SVD-based LoRA merging methods. However, on held-out and zero-shot tasks, uniform merging achieves competitive—and in some cases even better—results, suggesting that improving the generalization ability of LoRA merging methods remains an open challenge. Moreover, our current evaluation is limited to at most 10 tasks. Scaling to a larger number of tasks presents additional challenges for LoRA merging, which we plan to investigate in future work.

**Use of large language models statement** We use the LLM to polish the writing. All other parts, including experimental results, analyses were written by the authors and carefully verified for accuracy before and after any LLM-assisted editing.

## A.2 Skill Composition Tasks

> **GSM8k vs GSM8k-hard**
>
> GSM8k: A robe takes 2 bolts of blue fiber and half that much white fiber. How many bolts in total does it take?
> Reference Answer: 18
> GSM8k-hard: A robe takes 2287720 bolts of blue fiber and half that much white fiber. How many bolts in total does it take?
> Reference Answer: 3,431,580

Figure 9: The comparison between GSM8k and GSM8k-hard. The GSM8K uses a larger number to replace the original number.

Prabhakar et al. (2024) have used LoRA merging to achieve the *skill composition*. Skill here refers to specific capabilities that the LLM needs for customization to downstream use cases. For example, achieving a high score in GSM8k benchmark Cobbe et al. (2021) needs good *commonsense* and *arithmetic* skills.

To explicitly examine how reducing task interference, we evaluate CP merging on hard math word problems. Compared to GSM-8K (Cobbe et al., 2021), GSM8K-hard (Gao et al., 2023) contains problems of similar types but with more complex arithmetic operations. Gao et al. (2023) propose a program-aided method in which an LLM first generates a program as an intermediate reasoning step and then executes it using a Python interpreter to obtain the final answer. This setting requires the LLM to be proficient in both mathematical reasoning (to analyze the problem) and coding (to translate the reasoning into executable code). Following the same experimental setting in Prabhakar et al. (2024), we train separate LoRA adapters for math and code skills.

### A.2.1 GSM8k-hard Results

As depicted in Tab. 6, we evaluate the accuracy scores on the GSM-8k hard benchmark across various models. The base model achieves a modest accuracy of 0.043, highlighting its limitations in tackling GSM-hard tasks effectively. By training a LoRA adapter with the MathQA instruction dataset, we enhance the model's mathematical capabilities. Similarly, training with the Alpaca-code dataset boosts the model's coding proficiency. Combining both MathQA and Alpaca-code datasets in a single adapter yields an accuracy of 0.1349. Merging the math and coding skills through LoRA merging results in an accuracy of 0.1296, demonstrating the potential to integrate these abilities. TSV Merging and Knots perform similarly in terms of accuracy, with scores of 0.1349 and 0.1344, respectively. However, TSV Merging generates fewer invalid codes (224) compared to Knots (263), suggesting it is slightly more stable in this regard. Among the competing baselines, our proposed CP merging achieves the highest accuracy of 0.1569. Additionally, we

| Model | ACC | Invalid Code |
|---|---|---|
| **BASE** | 0.0430 | 524 |
| **LoRA(math)** | 0.1175 | 562 |
| **LoRA(code)** | 0.0796 | 159 |
| **Multi-task** | 0.1349 | 413 |
| **Uniform** | 0.1296 | 217 |
| **Ties-merging** | 0.1060 | 223 |
| **Task Arithmetic** | 0.1190 | 232 |
| **Ties w/DARE** | 0.1162 | 227 |
| **TA w/DARE** | 0.1200 | 247 |
| **ISO Merging** | 0.0840 | 394 |
| **TSV Merging** | 0.1349 | 224 |
| **Knots** | 0.1344 | 263 |
| **CP merging** | **0.1569** | 201 |

Table 6: Evaluation results on MATH-hard tasks. "Invalid" is the number of invalid codes that can not be executed by Python.

assess the "invalid code" metrics across methods. Notably, the LoRA code approach reduces invalid code instances from 524 in the base model to 159, indicating improved code quality.

The results demonstrate that our CP merging effectively reduces task interference by integrating math and coding skills, as evidenced by the lower number of invalid codes (201) compared to the base model (524) and other methods. In conclusion, CP merging not only achieves the highest accuracy but also maintains a low rate of invalid code generation, clearly demonstrating its superior performance in handling complex math-hard tasks.

### A.3  SVD illustration of U and V

The similarities in the $U$ and $V$ matrices can reflect correlations among different experts. To empirically investigate whether SVD fails to capture inter-task interference, we compute the pairwise cosine similarity of the $U$ and $V$ matrices across five diverse task experts (Figure 10). As observed in the heatmaps, the off-diagonal similarity scores are consistently low, suggesting that SVD factors are highly task-specific and lack a common orientation. Specifically, the $V$ matrix similarity (right) shows that the input-side features of different tasks are nearly orthogonal. This confirms that a naive merge of SVD-based adapters would result in information conflict, as the model would attempt to project different task logic onto misaligned subspaces.

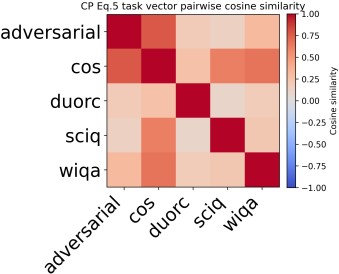 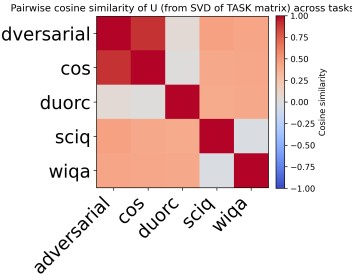 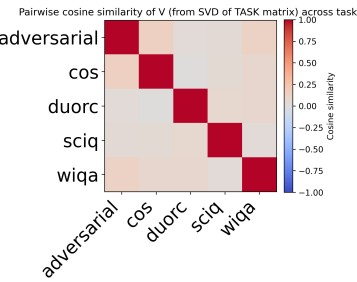

Figure 10: (left): pairwise cosine similarity between task experts using CP Eq.5. Each cell compares the task matrices $\Delta_i, \Delta_j$, highlighting cross-task dependence captured by CP's tri-factor interaction. (2)(mid) similarity between the $U$ matrix in SVD decomposition. (3) similarity between the $V$ matrix in SVD. We use layer 0 in the Phi-3 model and compute the cosine similarity across 5 experts.

We added a quantitative cross-task dependence analysis using the off-diagonal entries of pairwise expert-similarity matrices. Specifically, we compare CP-based task representations (Eq. 5 reconstruction) against SVD-based representations (U, V, and their mean). On the tested layer, CP-vs-SVD correlations are only moderate (Pearson: 0.578 vs U, 0.486 vs V, 0.576 vs (U+V)/2; Spearman: 0.305, 0.353, 0.377), and the average absolute discrepancy is non-trivial (0.204). This indicates that SVD and CP induce meaningfully different cross-task dependence structures rather than being equivalent. We also observe that SVD dependence is imbalanced across factors (mean off-diagonal: U=0.348, V=0.068), while CP task vector captures a coherent dependence pattern at the full task-matrix level (0.349). These quantitative results strengthen our motivation for CPD: it models cross-task interactions through a shared tri-factor structure and preserves dependence information beyond what is captured by SVD basis components alone.

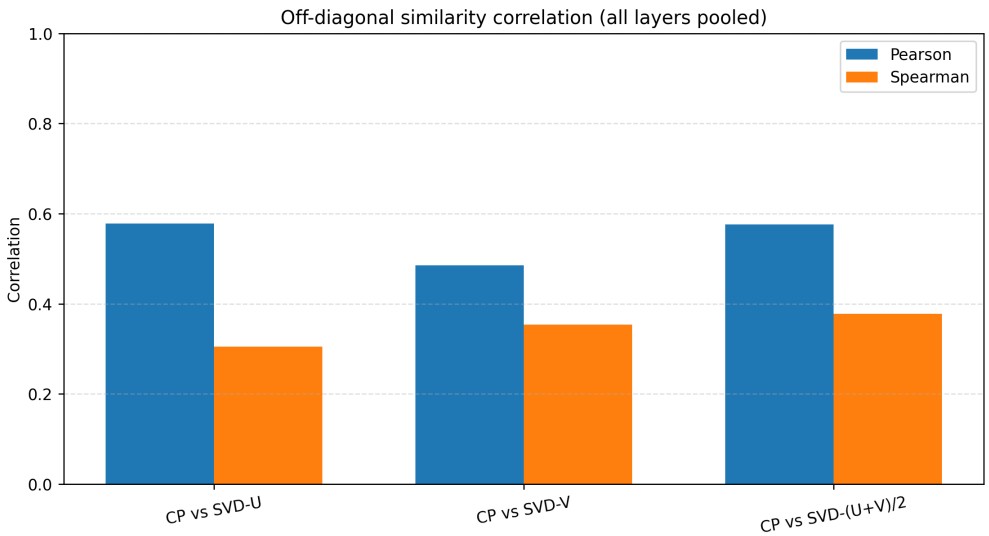

Figure 11: Quantitative cross-task dependence analysis using the off-diagonal entries of pairwise expert-similarity matrices.

### A.4 Experimental Setting for Flan tasks and Zero-shot downstream tasks.

### A.4.1 Implementation Details

For the CP decomposition in our merging algorithm, we use the open-source Python library Tensorly [1]. CP decomposition is solved using alternating least squares (ALS) via TensorLy's parafac solver (random initialization, fixed rank $R$, stopping with tolerance *tol* or maximum iterations *n-iter-max*).

To provide convergence evidence, we now log the reconstruction error at each ALS iteration and report the corresponding reconstruction MSE between the original tensor X and reconstructed tensor $\hat{X}$ : $\mathrm{MSE}_t = ||X - \hat{X}_t||_F^2/|X|$ (equivalently computed from TensorLy's relative reconstruction error trajectory). We include both per-layer convergence curves and an averaged summary across layers (mean $\pm$ std over layers), and observe stable monotonic error reduction with saturation after a modest number of iterations. These results support that the CPD optimization in Step 5 is numerically well-behaved and convergent in our setting.

### A.4.2 Datasets and Running Environment Setting

To enhance the LLM with various abilities for the above tasks, we construct a collection of LoRA adapters on diverse tasks. We use the FlanV2, an instruction-tuning dataset designed to scale both task diversity and model size, which has been shown to improve model performance significantly (Chung et al., 2024). We train each LoRA adapter independently, enabling it to specialize in a specific task. Finally, we select 256

---

[1]https://tensorly.org/dev/modules/generated/tensorly.decomposition.parafac.html

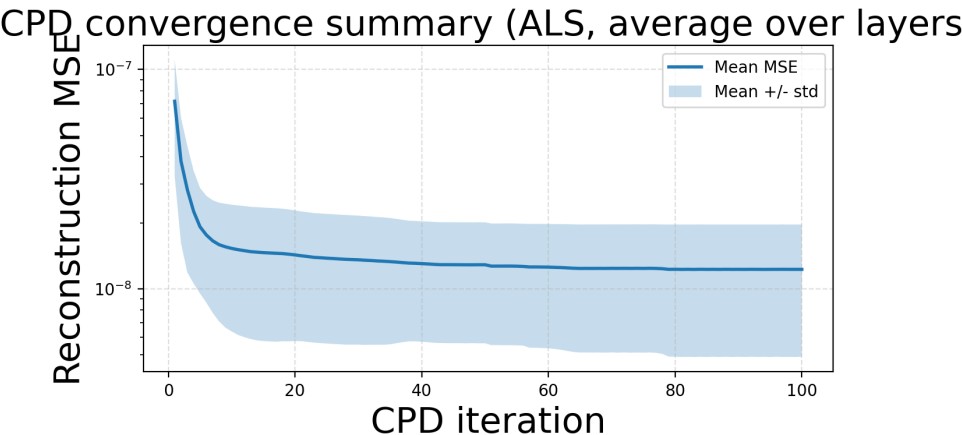

Figure 12: The CP decomposition based on Phi-3 on LoRA library with 5 experts.

tasks (Longpre et al., 2023), with the majority derived from P3 (Bach et al., 2022) and Flan2021 (Wei et al., 2021).

To evaluate the effectiveness of our approach across different backbone models, we conduct experiments using Phi-3 (Abdin et al., 2024), and Mistral-7B(mistralai/Mistral-7B-v0.1) (Jiang et al., 2023). In all cases, we apply LoRA adapters exclusively to the attention layers. Unless stated otherwise, our multi-task training and single-task adaptation scenarios employ a LoRA rank of 4, a dropout rate of 0.05, and a learning rate of 1e-4. Each LoRA adapter is trained for 5 epochs on either an H100 or A100 GPU. For the clustering step, we use the `sentence-transformers/sentence-t5-xxl` model for encoding. Given the typically large size of the dataset, we sample 20% of the data to train the k-means algorithm and then use the resulting clusters to predict labels for all samples.

### A.4.3 10 Held-in and 3 Held out task names

There are 256 pre-trained LoRA adapters used in Ostapenko et al. (2024). We randomly select 10 tasks as the held-in tasks and 3 tasks as the held-out tasks, followed by **?**.

**Held-in tasks**

- wiqa_what_is_the_final_step_of_the_following_process

- sciq_Multiple_Choice

- adversarial_qa_droberta_answer_the_following_q

- duorc_SelfRC_question_answering

- cos_e_v1_11_description_question_option_id

- wiki_qa_Is_This_True_

- quail_description_context_question_text

- wiki_hop_original_explain_relation

- duorc_ParaphraseRC_build_story_around_qa

- yelp_polarity_reviews_0_2_0

**3 held-out tasks**

- glue_sst2_2_0_0

- dream_read_the_following_conversation_and_answer_the_question

- race_middle_Read_the_article_and_answer_the_question_no_option_

## A.5  Cross layer analysis.

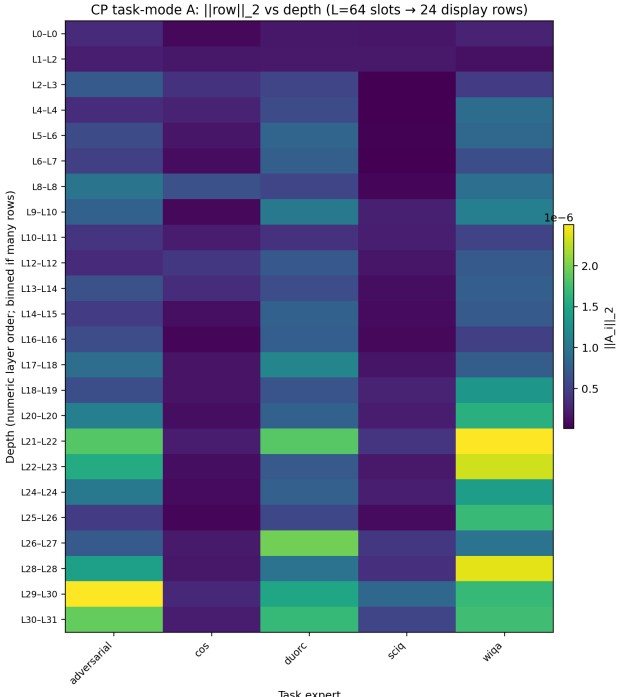

Figure 13: L2 norm of each task row vs layer.

We summarize each task's CP mixture at a layer by the L2 norm of its row in A and the entropy of the magnitude-normalized mixture (how peaked vs diffuse the rank weights are). As shown in the Figure 13. The heatmap shows that task influence is not uniform across depth: some tasks remain small across most layer–projection slots, while others concentrate in later blocks (notably deeper o_proj / qkv_proj rows), indicating that the learned decomposition reflects structured depth dependence rather than i.i.d. local noise. This directly addresses whether "importance" is constant: it is layer- and projection-dependent, with interpretable bands rather than flat profiles.

In Figure 14, we measure the cosine similarity between its A row at layer L and at the next processed layer, then report the mean ± std across tasks for each adjacent pair. The resulting curve is mixed: several transitions show positive mean alignment (e.g. L10 o_proj → L10 qkv_proj at 0.535 mean cosine, L27 qkv_proj → L28 o_proj at 0.454), while others are near zero or negative with large standard deviations (often 0.3–0.9), reflecting that (i) o_proj and qkv_proj are different operators with different induced geometry, and (ii) CP factors have scale/sign/permutation non-identifiability, so A is not guaranteed to be pointwise stable under independent per-layer fits. We therefore interpret this plot as a sanity check for continuity under our layer-wise fitting protocol, not as a claim of a single global latent "task embedding" shared verbatim at every depth.

Because CP rank order is arbitrary within each layer, we report Hungarian alignment of normalized factor columns between consecutive transformer indices within the same projection family (o_proj vs qkv_proj).

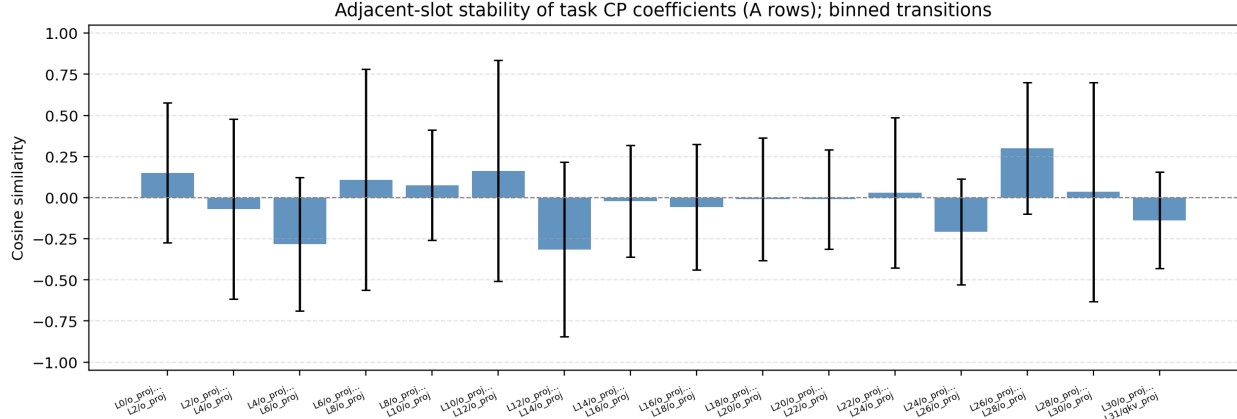

Figure 14: Adjacent-slot stability of task CP coefficients (A rows).

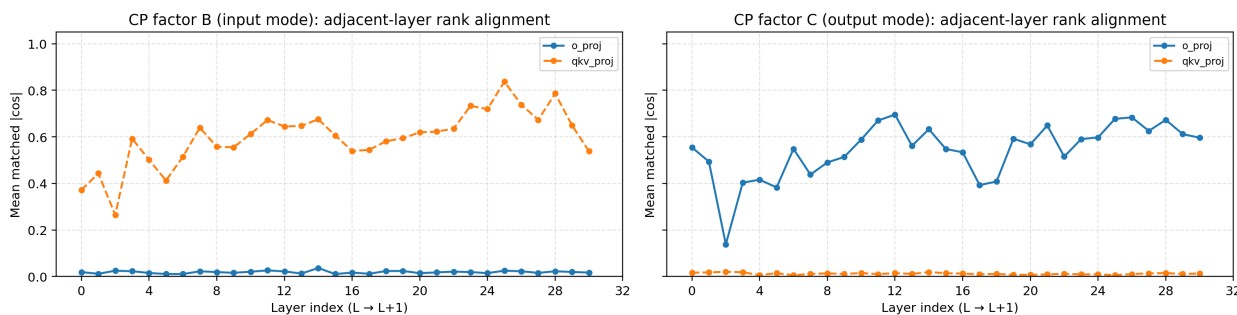

Figure 15: Rank-1 feature across depth. |cos| means the absolute value of the cosine similarity between two column vectors of the CP factor matrices B or C (after each column is L2-normalized).

As shown in the Figure 15, the results are split by factor and projection: for B on qkv_proj, matched |cos| between adjacent layers is typically moderate to high (often 0.37–0.84, e.g. peak ~0.84 at L25→L26), suggesting recurring rank-1 directions in the input mode along depth for QKV adapters. For B on o_proj, matched |cos| is much lower (often ~0.01–0.04), indicating that rank labels are not vertically stable for output projections under independent CPD—consistent with known CP identifiability limits. For C, o_proj shows higher adjacent matched similarity than qkv_proj (e.g. many o_proj pairs in the 0.4–0.7 range vs qkv_proj often ~0.006–0.02), so "vertical semantics" of rank-1 components must be stated separately for each factor and each projection type.

## A.6 Clusterized LoRA Library (C-LoRA)

As noted in Ostapenko et al. (2024), LoRA similarity can contribute to positive transfer. Their approach clusters tasks based on LoRA similarity, requiring the *task ID* for each example and the pre-training of a collection of LoRA adapters before clustering. In contrast, we argue that text-level clustering is more convenient while achieving competitive results. As shown in Fig. 16, we select 100 examples per cluster and visualize their embeddings. The embeddings reveal that similar texts are grouped into the same clusters, demonstrating the quality of our embedding representations.

To mitigate task conflicts at the *text-level*, we train each *expert* on a collection of similar tasks. This is achieved by clustering instructions and partitioning the training data into multiple clusters without requiring manual intervention. Clustering input texts into several groups can reduce task interference by allowing the model to adapt its task adapters or representations to distinct subsets of data with similar characteristics.

Formally, let $E(\cdot)$ denote a pretrained sentence encoder. For each data sample, the sentence representation of an instruction $I_i$ is computed as $e_i = E(I_i)$. We then apply the k-means algorithm to cluster all instruction embeddings $\{e_i\}$ in the training dataset into $K$ clusters. We train each LoRA adapter based on the instructions from each cluster. For each pre-trained weight $W_0$, we obtain corresponding LoRA adapters $\{A_i B_i, ..., A_K B_K\}$.

### A.6.1 Clustering to reduce the task interference at the text-level

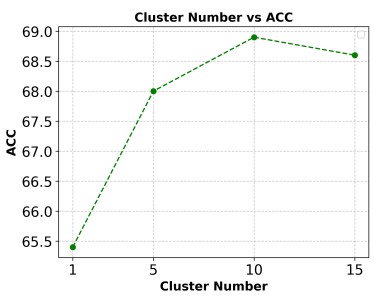 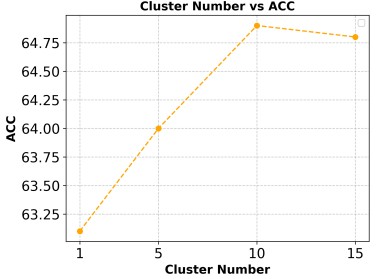 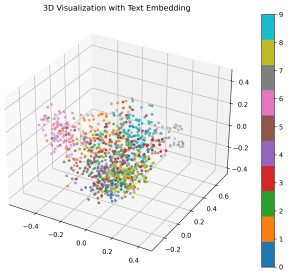

Figure 16: (left). The number of clusters and accuracy (ACC) using Phi-3 3B. Cluster analysis. The number of clusters and accuracy (ACC) using Mistral 7B. Clustering based on the sentence embedding. We visualize 100 samples in each cluster.

The Fig. 16 shows the relationship between the number of clusters and accuracy (ACC) using two different model backbones: Phi-3 3B (left) and Mistral 7B (right). For the Phi-3 3B backbone, accuracy increases from approximately 65.4% at 1 cluster to a peak of 68.9% at 10 clusters, stabilizing thereafter up to 15 clusters. In contrast, the Mistral 7B Backbone shows a more gradual improvement, rising from 63.1% at 1 cluster to 64.9% at 10 clusters. We think it is still a challenge to determine the ideal number of clusters. Further investigation with more sophisticated clustering techniques could provide deeper insights into the relationship between performance and the number of clusters.

### A.7 Zero-shot Results on Mistral 7B

For the Mistral (7B) model, the multi-task version achieves a notable improvement, raising the accuracy from 59.5% to 63.1%.

| | Common Sense | | | Question Answering | | | | Coding | | Reasoning | AVG |
|---|---|---|---|---|---|---|---|---|---|---|---|
| | Piqa | Wg | Hswag | Boolq | Obqa | ArcE | ArcC | HE | Mbpp | BBH | AVG |
| Mistral (7B) with LoRA library | | | | | | | | | | | |
| BASE | 81.1 | 66.5 | 78.8 | 82.2 | 44.6 | 68.9 | 49.6 | 28.0 | 47.5 | 47.9 | 59.5 |
| Multi-task | 81.9 | 68.6 | 79.5 | 84.6 | **50.4** | 84.8 | 60.0 | 24.4 | 47.5 | 49.2 | 63.1 |
| Uniform | 82.1 | 67.2 | 79.6 | 82.7 | 45.2 | 78.7 | 54.8 | 29.9 | 49.4 | 49.0 | 61.9 |
| Ties-merging | 82.4 | 70.6 | 79.9 | 87.6 | 44.2 | 81.1 | 57.3 | **31.7** | 47.9 | 46.2 | 62.9 |
| Task Arithmetic | 82.3 | 67.1 | 79.5 | 82.4 | 45.3 | 78.4 | 54.8 | 30.0 | 49.4 | 49.1 | 61.8 |
| Ties w/DARE | 82.3 | 70.3 | 79.8 | 87.4 | 44.3 | 81.3 | 57.2 | 31.6 | 47.6 | 46.3 | 62.8 |
| TA w/DARE | 82.2 | 67.3 | 79.5 | 82.2 | 45.4 | 78.3 | 54.9 | 30.1 | 49.5 | 49.2 | 61.9 |
| TSV merging | 81.6 | 70.3 | 78.9 | 82.8 | 42.8 | 84.0 | 58.3 | 28.7 | 49.7 | 49.5 | 62.7 |
| Mistral (7B) with C-LoRA library | | | | | | | | | | | |
| Uniform | 82.4 | 71.4 | 81.2 | 87.6 | 49.2 | 86.0 | 60.5 | 29.3 | 50.0 | 50.0 | 64.9 ↑2.2 |
| CP-merging | **82.6** | **71.7** | **81.6** | **87.8** | 49.3 | **86.0** | **60.6** | 29.3 | **50.2** | **50.7** | **65.0** ↑2.3 |

Table 7: 10 downstream Zero-shot results based on Mistral 7B.

