# OpenReview forum: "CP Merging: Joint LoRA Merging using Canonical Polyadic Decomposition"
_TMLR — Under review for TMLR_

### Review · Reviewer_5xLb · 2025-12-21

**Summary Of Contributions:**

The paper studies multi-task LoRA adapter merging, where each LoRA adapter is fine-tuned on a different task and needs to be merged into a single adapter for multi-task inference. Different from previous approaches that use SVD-based methods for merging, the paper investigates using CP decomposition instead. Experiments show strong performance compared with state-of-the-art methods.

**Audience:**

Yes

**Audience Explanation:**

The idea of using CP decomposition for adapter merging is an interesting alternative to SVD. The paper clearly shows the implementation and explains how the decomposed tensor factors represent different dimensions of the original merged adapter tensor. Readers may be interested in further research on using tensor-related methods to improve merging performance

**Broader Impact Concerns:**

I don't think there is any ethical concerns for this paper.

**Claims And Evidence:**

Yes

**Claims Explanation:**

The claims are generally clear, with sufficient ablation studies and experimental results. See the requested changes for potential improvements to this paper.

**Requested Changes:**

- It would be great if the authors could provide a preliminary section, before the introduction, on SVD-based adapter merging with the basic adapter merging setup. I am not familiar with this direction, and it took me some time to understand the basic LoRA adapter merging formulation.
- I am wondering whether the proposed method is limited to LoRA adapter merging. Could the authors test it on more recent PEFT adapters, such as DoRA [1] or recent tensorized PEFT adapters [2][3]? Demonstrating effectiveness across different PEFT methods would further enhance the applicability of the proposed approach.
- For the results in Section 6.5, I am curious about why CP merging is reported to be 10× faster than SVD-based methods such as TSV merging. Intuitively, SVD-based methods are often faster than tensor decompositions (e.g., those using alternating least squares), and some discussion or analysis would be helpful.

[1] Liu, Shih-Yang, et al. "Dora: Weight-decomposed low-rank adaptation." Forty-first International Conference on Machine Learning. 2024.
[2] Yang, Yifan, et al. "Loretta: Low-rank economic tensor-train adaptation for ultra-low-parameter fine-tuning of large language models." arXiv preprint arXiv:2402.11417 (2024).
[3] Bershatsky, Daniel, et al. "LoTR: Low tensor rank weight adaptation." arXiv preprint arXiv:2402.01376 (2024).

---

### Review · Reviewer_Qf7m · 2026-04-21

**Summary Of Contributions:**

The paper presents CP Merging, a framework that treats multi-task LoRA merging as a joint tensor decomposition problem. By stacking $N$ adapters into a third-order tensor ($R^{N \times d \times k}$) and applying Canonical Polyadic (CP) Decomposition, the method disentangles task-specific coefficients ($a_r$) from a shared weight basis ($b_r$, $c_r$). This enables a "joint" merge that reduces task interference (sign/magnitude conflicts) more effectively than traditional 2D or 1D methods.

**Audience:**

Yes

**Audience Explanation:**

Yes. This is very relevant to researchers in parameter-efficient fine-tuning (PEFT) and model merging domain. As we scale toward thousands of task-specific adapters, the transition from simple averaging to principled tensor decomposition is a logical and necessary evolution for the field.

**Claims And Evidence:**

Yes

**Claims Explanation:**

The empirical evidence is robust, particularly on "Hard" variants of reasoning benchmarks where interference usually causes the most degradation. The comparison against Ties-Merging and DARE is clear and shows a consistent advantage.

**Requested Changes:**

Request of changes:

* Incremental Adapter Incorporation: The authors should address the "Open World" scenario where a new adapter is released after the initial merge. Currently, the third-order tensor requires a full re-calculation because the task dimension $N$ changes. A discussion or experiment on an incremental update strategy would be extremely valuable. For example, fixing the weight basis ( $b_r$, $c_r$) and solving only for the new task coefficients $a_{N+1, r}$. This is vital for the method's engineering viability in dynamic environments.

* Cross-Layer Coefficient Analysis: While merging is performed layer-by-layer, it is unclear if the "importance" of a task remains consistent throughout the model’s depth. An analysis of the merge coefficients across different layers is missing. Do certain rank-1 components represent the same semantic "features" in Layer 1 vs in Layer 32? Understanding this vertical consistency would clarify if the decomposition is capturing global task logic or merely local weight patterns.

* Categorical Distinction in Literature Review: Explicitly distinguish between dense model merging (e.g., Fisher, Ties) and low-rank (LoRA) merging to highlight why traditional methods struggle with the unique alignment challenges of LoRA subspaces.

---

### Review · Reviewer_PbLM · 2026-04-22

**Summary Of Contributions:**

**Summary**

The paper proposes a method for merging multi-task LoRA adapters by stacking these adapter layers into a tensor $\mathcal{T} \in \mathbb{R}^{d_{in}\times d_{out}\times N}$. The paper then applied CP-Decomposition to obtain full-rank factor matrices $A \in \mathbb{R}^{N\times R}$, $B\in \mathbb{R}^{d_{in}\times R}$, $C\in \mathbb{R}^{d_{out}\times R}$, and then reconstructs to an full $R$ rank matrix, which is then added to the pretrianed weights. Unlike prior work, which compute the SVD of each task adapter and concatenate the SVD basis, the proposed tensor formulation explicitly represents the tasks $N$ along a third additional dimension. Experimentally, the paper reports improved performance in held-in/out and zero-shot benchmarks.

**Strengths**
1. The proposed tensor formulation for multi-task LoRA merging is intuitive and novel. Specifically, how it introduces an explicit task dimension.
2. Empirical results show performance improvements on held-in/out and zero shot benchmarks.

**Weaknesses**
1. The paper states SVD basis captures intra-task specific transformations, but fails to capture inter/cross-task transformations. The paper does not show how the evaluated LLMs fail in this setting. A plot of cosine similarity between [U_1, U_2, U_3, ... U_N] and [V_1, V_2, V_3, ... V_N] separately, can highly strengthen the case for tensor formulation.
2. The choice of CP decomposition is not sufficiently supported. The statement that CP is a generalization of SVD to higher-order tensors is only partially true. Unlike SVD, CP factors are not guaranteed to be orthogonal, and the paper does not show how factor matrix columns are, for examples, is there sufficient diversity in matrix $A$? Moreover, Is other orthogonal deterministic tensor decompositions like HOSVD considered?
3. The paper do not present convergence evidence for CPD step in Algorithm 1 Step 5. Please specify the CPD optimisation algorithm used like ALS, NLS, or gradient based. Please report MSE between original and reconstructed tensor over iterations.
4. Given that CP full rank is NP hard [1] to compute, reporting rank against reconstructed mse would establish the chosen rank and provide an estimate on approximation quality.
5. The CP-STI formulation needs important clarification. Prior STI scores rely on orthonormal SVD factors so the term $U^TU-I$ and $V^TV-I$ removes diagonal terms. Whereas CP factors are not generally orthonormal, so the $A^TA-I$ still contains large diagonal components that dominate the L1 norm. A normalised factor matrix should be considered to isolate the off-diagonal elements like STI.
6. Please clarify whether CPD (step 5) time is included in CPMerging cost in Table 4. Moreover, if wall clock time is reported as computation cost, please also provide GFLOPs or synchronised GPU time, because wall time is dependent on multiple external factors.

**Audience:**

Yes

**Audience Explanation:**

I believe the paper will likely be of interest to the community. As the PEFT methods are becoming more efficient, the task of combining PEFT adapters for multi-task adaptation using tensor formulation is a relevant direction.

**Broader Impact Concerns:**

The paper does not explicitly provide broader impact. Based on the current scope, no ethical concerns are present.

**Claims And Evidence:**

No

**Claims Explanation:**

1. The paper states that SVD basis prevents identification of optimal orthogonal bases, but donot clearly demonstrate that drop in performance for SVD methods in Tables 1 and 2 is because of the cross-task interference.

2. The claim that CP based factorisation reduces cross task interference and preserve intra task relevant information is not established. Neither the CP factors are orthonormal by design nor the task factor matrix $A$ is provided.

3. The proposed CP-STI can help understand task interference in factor matrices, but it assumes CP factor matrices to be orthonormal. Unless orthogonal CPD algorithm with normalised factors is used, it is not guaranteed in its current form and implementation details of CPD algorithm are missing.

**Requested Changes:**

1. Please consider citing the original Tucker [2] and CPD [3, 4] works in Section 3.1

2. Strengthen the motivation for CPD over the SVD basis by showing higher cross-task dependence on the SVD basis, and CPD factor matrix A showing lower dependence.

3. Add implementation details for CPD of $\mathcal{T}$, including the algorithm used. Provide convergence plots with multiple seeds, as these methods are generally initialisation dependent.

4. Following weakness 5, please explain why L1 norm of CP-STI is helpful and why not Frobenius norm which measures matrix energy. Additionally please address diagonal dominance in $A^TA-I$.

5. Please clarify the computation cost measurement as stated above and provide the libraries used.

6. Equation 1 corresponds to normalised CP variant with $\lambda$'s [1], while the methodology appears to use non-normalised version. Please make the equation consistent with methodology.

7. Please update the references, there are few works like Zhang et al 2023a, Hellaswag, Bitfit, Dora and more are published.

[2] Tucker, Ledyard R. "Implications of factor analysis of three-way matrices for measurement of change." Problems in measuring change 15.122-137 (1963): 3.

[3] Hitchcock, F. L. The expression of a tensor or a polyadic as a sum of products. Journal of Mathematics and Physics,
6(1-4):164–189, 1927.

[4] Hitchcock, F. L. Multiple invariants and generalized rank
of a p-way matrix or tensor. Journal of Mathematics and
Physics, 7(1-4):39–79, 1928.